# Relationship between Changes in Foot Arch and Sex Differences during the Menstrual Cycle

**DOI:** 10.3390/ijerph20010509

**Published:** 2022-12-28

**Authors:** Mutsuaki Edama, Tae Ohya, Sae Maruyama, Mayuu Shagawa, Chie Sekine, Ryo Hirabayashi, Hirotake Yokota, Tomonobu Ishigaki, Hiroshi Akuzawa, Ryoya Togashi, Yuki Yamada, Tomoya Takabayashi

**Affiliations:** 1Athlete Support Medical Center, Niigata University of Health and Welfare, Niigata 950-3198, Japan; 2Institute for Human Movement and Medical Sciences, Niigata University of Health and Welfare, Niigata 950-3198, Japan

**Keywords:** menstrual cycle, sex difference, arch height index, arch height flexibility

## Abstract

This study investigated the relationship between changes in foot characteristics and sex differences during the menstrual cycle in healthy male and female university students. We examined 10 female subjects and 14 male subjects. The menstrual cycle was divided into the three phases: the early follicular phase, ovulatory phase, and luteal phase via basal body temperature, an ovulation kit, and salivary estradiol and progesterone concentration measurements. Foot characteristics required for the calculation of the arch height index (AHI) were measured using a three-dimensional foot scanner under conditions of 10% and 50% weight-bearing loads. Arch height at 50% of foot length and truncated foot length were measured, and AHI was calculated by dividing arch height by truncated foot length. Arch height flexibility (AHF) was defined as the change in arch height from 10% weight-bearing load to 50% weight-bearing load. AHI was significantly lower in females than in males in the early follicular and ovulatory phases but did not differ significantly between males and females in each phase. AHF did not differ significantly between males and females in each phase. AHI and AHF showed no periodic fluctuation, suggesting that sex differences in AHF may be absent.

## 1. Introduction

The frequency of sports injuries is reportedly higher in women than in men [1,2]. A lateral ankle sprain is one of the most common injuries resulting from recreational and competitive sports activities [3] and reportedly occurs more frequently in women than in men [4]. For that reason, a relationship between the menstrual cycle and sports injury has been suggested [1,2].

The menstrual cycle is regulated by the female hormones, estradiol (E2) and progesterone (P4), and is mainly divided into the follicular, ovulatory, and luteal phases [5,6]. E2 peaks during the ovulatory phase and P4 peaks during the luteal phase. Regarding the knee joint, receptors for E2 and P4, a type of estrogen, are reportedly present in the anterior cruciate ligament (ACL) of the human knee [7]. The ACL possesses receptors for E2 and P4 [7], and in vitro studies of the ACL have shown that increasing the concentration of E2, as a form of estrogen, causes a decrease in human ACL fibroblast proliferation and Type I procollagen synthesis [8]. On the other hand, the concentration of P4 decreases the effects of an increased concentration of E2 on ACL tissue metabolism [9]. Thus, fluctuations in female hormones clearly affect tissue metabolism in the ACL. On the other hand, the presence of E2 and P4 receptors in the tissue structures of the foot and ankle joints has not been clarified. However, previous studies have investigated female hormone levels in relation to plantar fascia elasticity and reported that plantar fascia elasticity increases during ovulation, when estrogen levels are maximal [10,11]. Conversely, a previous study [11] that investigated the rate of anterior talofibular ligament lengthening during the menstrual cycle found no cyclic variation. A clear consensus on the effect of female hormones on foot posture thus remains lacking.

Abnormal foot postures, such as pes planus and cavus feet, are often linked to musculoskeletal disorders, and the assessment of foot posture is essential for determining the possible risk factors and causes of musculoskeletal disorders [12]. Among static foot assessments, the arch height index (AHI) is a method of evaluation for classifying pes planus and cavus feet and has consistently shown excellent intra- and inter-test reliability [13,14]. AHI has been validated by previous studies in comparison with radiographic measurement [15] and offers strong predictive ability regarding dynamic foot postures, such as walking and jogging [16]. In addition to AHI, arch height flexibility (AHF) is a useful evaluation method for classifying flexible and stiff feet [13]. The correlation between arch structure and injury may be related to the fact that foot structure influences foot function. Foot structure is often defined by arch height, although arch flexibility may be just as important to form a more complete description [17]. A previous study found that runners with high arch structure but differing arch mobility exhibited differences in select lower extremity movement patterns and forces [18]. Evaluating AHI and AHF is, thus, considered useful for clarifying the relationship between foot misalignment and foot injury. A previous study [12] that evaluated foot posture using AHI and AHF reported that AHI was significantly lower in women than in men under all loading conditions (10%, 50%, and 90%), while AHF decreased with increasing loading in both sexes [12]. However, the effects of fluctuations in female hormones during the menstrual cycle on changes in AHI and AHF remain unclear.

The present study was conducted to clarify the relationship between foot posture changes and sex differences during the menstrual cycle (early follicular phase, ovulatory phase, and luteal phase) in healthy male and female university students. We hypothesized that AHF would be significantly higher in the ovulatory and luteal phases of the menstrual cycle in females than in the early follicular phase.

## 2. Materials and Methods

### 2.1. Subjects

For this study, 67 female university students were interviewed and given a questionnaire to determine if they met the study criteria. Inclusion criteria were as follows: (1) a regular menstrual cycle with a length of 25–38 days [19]; (2) biphasic basal body temperature [20]; (3) no orthopedic abnormalities or injuries to the foot and ankle joint; (4) no use of oral contraceptives or other hormonal agents within the past 6 months [21]; and (5) no current exercise habits of more than twice a week [22]. Ten female subjects (mean age, 21.1 ± 0.7 years; height, 159.7 ± 4.7 cm; weight, 50.3 ± 7.7 kg; cycle length, 31.1 ± 1.8 days) met inclusion criteria 1–4 and agreed to participate in the study (Figure 1). Fourteen male subjects (mean age, 21.9 ± 1.2 years; height, 170.8 ± 4.6 cm; weight, 62.7 ± 7.1 kg) met inclusion criteria 2 and 3 and agreed to participate in the study (Figure 2). The study was performed according to the Declaration of Helsinki, after receiving approval from the ethics committee at our institution (approval no. 17946). The study content was fully explained to each subject and written down; informed consent was obtained from all subjects before their participation in the study.

### 2.2. Recording the Menstrual Cycle

Using a basal body thermometer (Citizen Electronic Thermometer CTEB503L; Citizen Systems Co., Tokyo, Japan), subjects were asked to measure basal body temperature upon waking up every morning from about 2 months before the start of the experiment. To estimate the day of ovulation, subjects were given an ovulation kit (Doctor’s Choice One-Step Ovulation Test Clear: Beauty and Health Research, Torrance, CA, USA). Daily basal body temperature and menstrual period were recorded. The results of the ovulation kit were recorded in the conditioning management system (ONE-TAP SPORTS, Euphoria Co., Tokyo, Japan). As for the transition to the high-temperature phase, we judged that basal body temperature shifted from the lower to the higher temperature phase and appeared to be biphasic when basal body temperature on 3 consecutive days after the estimated day of ovulation was ≥0.2 °C higher than the average basal body temperature during the 6 successive days before the estimated day of ovulation [20].

### 2.3. Timing of Measurements

E2 and P4 concentrations and the 3D measurements of the foot were determined twice each during the early follicular, ovulatory, and luteal phases, for a total of six times. The phases were defined as follows: early follicular phase, 3–4 days after the start of menstruation; ovulation phase, 2–4 days after the day on which the ovulation kit gave a positive result; luteal phase, 5–10 days after the start of the high-temperature phase. In men, to provide measurement intervals corresponding to the phases of the menstrual cycle, the date of the starting measurement was set as the first day, with phase 1 then defined as days 1 and 2, phase 2 as days 15 and 16, and phase 3 as days 22 and 23. To account for diurnal variations, measurements were taken between 08:00 and 12:00. The temperature in the room was set at 20–25 °C.

### 2.4. Measurement Methods

Using a saliva collection kit (SalivaBio A; Salimetrics, Carlsbad, CA, USA), E2 and P4 concentrations were measured. Subjects strictly obeyed the following points before saliva collection to avoid possible influences on E2 and P4 concentrations: (1) food intake restrictions within 60 min; (2) dairy product restrictions within 20 min; (3) alcohol restrictions within 12 h; (4) sugary, acidic, or caffeinated drink restrictions within 20 min; and (6) saliva collection restrictions during the 48 h after dental treatment. Additionally, participants were instructed to rinse their mouths before the experiment began, to remove any food particles. Saliva samples were also taken more than 10 min after mouthwash to prevent decreases in E2 concentration. A straw (Siva Collection Aid; SAL) was used to collect saliva in the mouth for one minute before it was drained into a saliva collection container (Cryovial; SAL). Immediately following collection, the saliva sample was frozen in a freezer at below −80 °C. After gathering all samples, the Funakoshi Corporation (Tokyo, Japan) was entrusted with the analysis of E2 concentrations. The High Sensitivity Salivary 17-Estradiol Enzyme Immunoassay Kit (SALIMETRICS) was used to measure the E2 concentrations. Samples were thawed at room temperature, mixed by vertexing, centrifuged at 1500× *g* for 15 min, and then analyzed using an enzyme-linked immunosorbent assay. The dilutions were all consistently onefold (undiluted solution) [23,24].

For the measurement of general characteristics and foot arch height, all subjects underwent the measurement of their height and weight using a digital height meter (AD6400; A&D, Tokyo, Japan) and body composition analyzer (BC-118D; TANITA, Tokyo, Japan), respectively. The foot characteristics required for the calculation of AHI were measured using a 3-dimensional foot scanner (Dream GP, Osaka, Japan). Foot measurements were taken under two conditions: 10% and 50% weight-bearing (WB) loads (Figure 3). When the foot scanner is activated, the scanner’s laser rotates on the rail around the foot, measuring about 30,000 positions, including the instep, heel, sole, and toes [25]. Participants stood with one foot inside the foot scanner and the other foot set on an adjacent platform next to and level with the platform inside the scanner. The assessor carefully ensured that the participant looked straight ahead and stood as still as possible, and the participant did not lean the body or bend at the hip and knee. Foot characteristics were measured by an experienced researcher. The arch height at 50% of foot length and truncated foot length (distance from the first metatarsophalangeal joint to the heel) were measured, and AHI was calculated by dividing the arch height by truncated foot length [15]. AHF was defined as the change in arch height from 10% WB load to 50% WB load (10%–50% AHF). According to a previous study [17], AHF was calculated using the following equation:(1)$$10\%-50\%\;\mathrm{AHF}=\frac{\mathrm{arch}\;\mathrm{height}\;\left(10\%\mathrm{WB}\right)-\mathrm{arch}\;\mathrm{height}\;\left(50\%\mathrm{WB}\right)}{0.4\times \mathrm{body}\;\mathrm{weight}}\times 100\left[\mathrm{mm}/\mathrm{kN}\right].$$

The foot characteristics required for the calculation of arch height index were measured using a 3-dimensional foot scanner. Foot measurements were taken under two conditions: 10% and 50% weight-bearing loads. Arch height at 50% of foot length and truncated foot length was measured, and the arch height index was calculated by dividing the arch height by truncated foot length. Arch height flexibility was defined as the change in arch height from 10% weight-bearing load to 50% weight-bearing load.

### 2.5. Intra-Rater Reliability

To assess the Intra-rater reliability of the AHI measurements, we recruited 5 adult males (mean age, 23.4 ± 0.8 years; height, 168.1 ± 5.7cm; weight, 60.9 ± 4.0 kg) and 5 adult women (mean age, 21 ± 0.9 years; height, 166.7 ± 8.6cm; weight, 59.3 ± 11.9 kg) who had left feet without orthopedic diseases or pain in the lower limbs. The measurement was repeated on 2 or more separate days within 1 week, and the intraclass correlation coefficient (ICC) (1,1) was calculated.

### 2.6. Statistical Analysis

The Shapiro–Wilk test was used to assess the normality of the different variables. The following analysis was used since normality was confirmed for all conditions. Intra-rater reliability was calculated with the ICC, interpreted according to the criteria of Landis and Koch [26]: poor, <0.00; slight, 0.00–0.20; fair, 0.21–0.40; moderate, 0.41–0.60; substantial, 0.61–0.80; and almost perfect, 0.81–1.00. A split-plot repeated-measures analysis of variance was used to compare body weight, E2, P4, 10% AHI, 50%AHI, and 50–10%AHF for each menstrual phase in each male and female. The Bonferroni method was used as a post hoc test. An unpaired *t*-test was used to compare E2, P4, 10%AHI, 50%AHI, and 50%–10%AHF for each menstrual phase among males and females. The level of significance was 5%. Statistical analyses were performed using SPSS version 24.0 (IBM Corp, Tokyo, Japan).

## 3. Results

The resulting ICC (1,1) was 0.934 for 10%AHI measurements and 0.973 for 50%AHI measurements. According to the criteria of Landis and Koch, [26] reproducibility is considered almost perfect for ICCs ≥ 0.81. The reproducibility of AHI measurement in this study was therefore considered high.

Changes in E2 and P4 concentrations during each phase are shown in Table 1. Female E2 concentrations were significantly higher in the ovulatory phase than in the early follicular phase (*p* = 0.047). Female P4 concentrations were significantly higher in the luteal phase than in the early follicular phase (*p* = 0.022). Female P4 concentrations were significantly higher in the luteal phase than in the male phase (*p* = 0.005). E2 and P4 concentrations in men did not differ significantly between phases (Table 1).

Changes in body weight, AHI, and AHF at each phase are shown in Table 2. No significant differences in body weight were noted between men and women in each phase. AHI (10% load condition) was significantly lower in females than in males in the early follicular and ovulatory phases (*p* = 0.032 and *p* = 0.024, respectively), but no significant difference was seen in AHI (50% load condition). There was no significant difference between menstrual cycles in both AHI. AHF did not differ significantly between males and females in each menstrual cycle (Table 2).

## 4. Discussion

The purpose of this study was to clarify the relationship between foot characteristics changes and sex differences during the menstrual cycle (early follicular phase, ovulatory phase, and luteal phase) in healthy college students. To the best of our knowledge, this represents the first study to clarify the relationship between foot characteristics (AHI and AHF) and changes in female hormones during the menstrual cycle.

The menstrual cycle is generally divided into four phases: the early follicular phase; late follicular phase; ovulatory phase; and luteal phase. However, the three phases of the early follicular phase (low E2 and low P4), ovulatory phase (high E2 and low P4), and luteal phase (high E2 and high P4) are considered important in relation to fluctuations in female hormones [6]. In the present study, female E2 concentrations were significantly higher in the ovulatory phase than in the early follicular phase, and P4 concentrations were significantly higher in the luteal phase than in the early follicular phase. The 10 female subjects recruited for this study thus had normal menstrual cycles and exhibited normal female hormone fluctuations.

In this study, the AHI (at 10% load condition) was significantly lower in females than in males in the early follicular and ovulatory phases, but no significant difference was seen in AHI (at 50% load condition). There was no significant difference between menstrual cycles in both AHIs. AHF did not differ significantly between males and females in each phase.

Previous studies have investigated AHIs in all load conditions (10%, 50%, and 90% of weight-bearing load) showed significant differences between the genders, and the AHI of female participants was significantly less than that of male participants [12]. Although the results were different from the results of this study, a previous study [12] used the arch height index measurement system to measure AHI, but in this study, a three-dimensional foot scanner was used. It was suggested that the difference might have an effect. In addition, previous studies have investigated female hormone levels in relation to plantar fascia elasticity and reported that plantar fascia elasticity increases during ovulation, when estrogen levels are maximal [10,11]. On the other hand, a previous study [11] that investigated the rate of anterior talofibular ligament lengthening during the menstrual cycle reported no cyclic variation. Thus, a clear, consistent view on the effect of female hormones on foot posture remains lacking. In this study, E2 concentrations were significantly higher in the ovulatory phase than in the early follicular phase. P4 concentration was significantly higher in the luteal phase than in the early and late follicular phases, suggesting that E2 and P4 concentrations showed normal cyclic variation. Although the presence or absence of E2 and P4 receptors in the foot tissue structure has not been clarified [10,11], AHI and AHF, which indicate foot characteristics in the menstrual cycle, may not be affected by E2 and P4 and may not fluctuate cyclically.

On the other hand, increases or decreases in body weight may influence the results. Premenstrual syndrome (PMS) is a cluster of physical and emotional symptoms that appear on a regular basis before the onset of menstrual bleeding [27]. Symptoms include bloating, a sense of increased body weight, breast pain, ankle swelling, irritability, aggressiveness, depression, lethargy, and food cravings [27]. In addition, no difference was apparent between the AHI values of men and women, although the arches in women were significantly less stiff [28]. These results suggest that women are more likely than men to be affected by weight gain or losses in AHI or AHF, and weight gain or loss in measurements over a long period of time may be expected to offset the results of AHI and AHF. However, since no change in body weight was seen during each phase, effects on weight change were considered lacking. In the future, the effect of foot swelling, as one symptom of PMS, on foot characteristics should also be examined.

A limitation of this study was the small sample size of 10 female subjects. However, the results were highly reliable because menstrual cycles were checked not only during the experiment, but also before the experiment, appropriate exclusion rules were implemented, and fluctuations in E2 and P4 concentrations were appropriate.

## 5. Conclusions

In this study, AHI was significantly lower in females than in males during the early follicular and ovulatory phases, but no significant differences between males and females were seen in each phase. In addition, AHI did not differ significantly between males and females at each phase. The AHI and AHF, therefore, showed no periodic fluctuations, suggesting that sex differences in AHF may be lacking.

## Figures and Tables

**Figure 1 ijerph-20-00509-f001:**
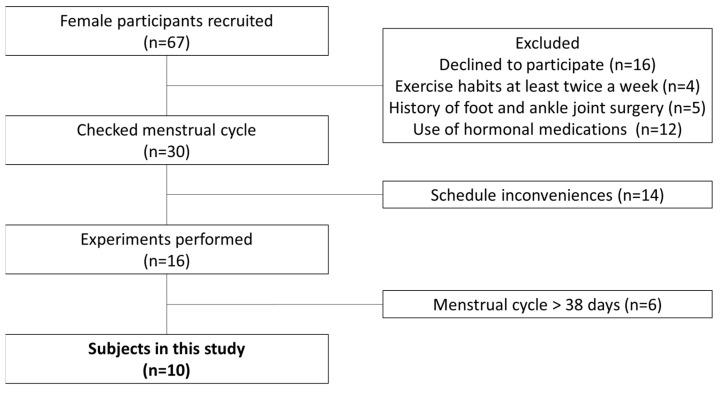
Flowchart for selection of female participants.

**Figure 2 ijerph-20-00509-f002:**
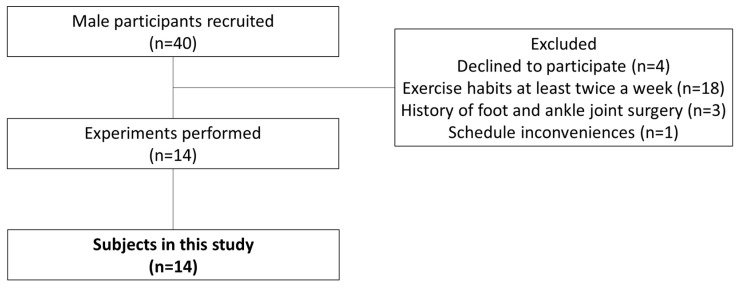
Flowchart for selection of male participants.

**Figure 3 ijerph-20-00509-f003:**
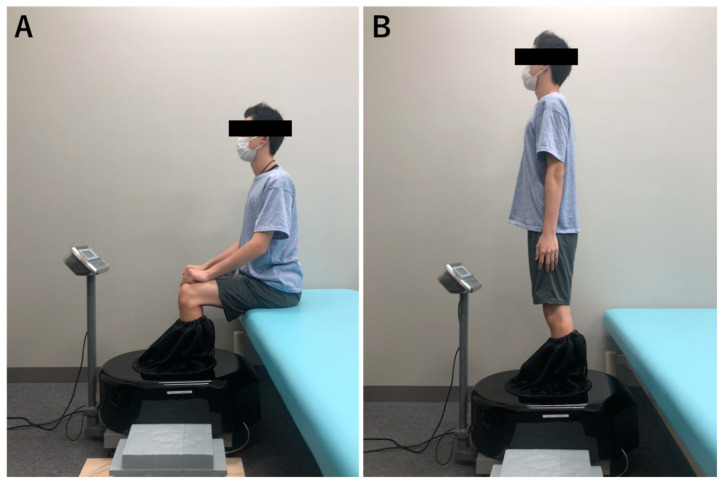
Methods of measuring foot characteristics. (**A**) Ten percent load condition. (**B**) Fifty percent load condition.

**Table 1 ijerph-20-00509-t001:** Changes in estradiol and progesterone concentrations during the menstrual cycle.

	Early Follicular Phase (Phase 1)	Ovulation Phase (Phase 2)	Luteal Phase (Phase 3)	Total
Estradiol [pg/mL]				
Male (*n* = 14)	1.28 ± 0.39	1.26 ± 0.30	1.30 ± 0.40	1.28 ± 0.36
Female (*n* = 10)	1.14 ± 0.14	1.49 ± 0.33 *^a^*	1.40 ± 0.34	1.35 ± 0.27
Progesterone [pg/mL]				
Male (*n* = 14)	149.77 ± 82.24	169.33 ± 83.99	163.77 ± 86.04	160.95 ± 84.09
Female (*n* = 10)	153.10 ± 74.97	208.15 ± 105.11	403.23 ± 284.38 *^b,c^*	254.83 ± 154.82

Values are presented as mean ± SD. *^a^* Statistically significant difference compared with the early follicular phase in females (*p* = 0.047). *^b^* Statistically significant difference compared with the early follicular phase in females (*p* = 0.022). *^c^* Statistically significant difference compared with the luteal phase in males (*p* = 0.005).

**Table 2 ijerph-20-00509-t002:** Changes in AHI and AHF during the menstrual cycle.

	Early Follicular Phase (Phase 1)	Ovulation Phase (Phase 2)	Luteal Phase (Phase 3)	Total
Body Weight (kg)				
Male (*n* = 14)	62.7 ± 7.3	62.8 ± 7.0	62.8 ± 7.0	62.7 ± 7.1
Female (*n* = 10)	50.4 ± 8.2	50.2 ± 7.8	50.4 ± 7.9	50.3 ± 8.0
AHI (10% load condition) (unitless)				
Male (*n* = 14)	0.38 ± 0.02 *^a^*	0.38 ± 0.02 ^*b*^	0.38 ± 0.01	0.38 ± 0.02
Female (*n* = 10)	0.37 ± 0.02	0.37 ± 0.02	0.37 ± 0.02	0.37 ± 0.02
AHI (50% lord condition) (unitless)				
Male (*n* = 14)	0.35 ± 0.01	0.35 ± 0.02	0.35 ± 0.02	0.35 ± 0.02
Female (*n* = 10)	0.34 ± 0.02	0.34 ± 0.02	0.34 ± 0.02	0.34 ± 0.02
10–50%AHF (mm/kN)				
Male (*n* = 14)	19.6 ± 4.3	20.9 ± 5.3	18.7 ± 5.4	19.7 ± 5.0
Female (*n* = 10)	19.2 ± 5.6	21.2 ± 5.2	22.4 ± 5.9	19.6 ± 4.3

Values are presented as mean ± SD. *^a^* Statistically significant difference compared with the early follicular phase in females (*p* = 0.032). *^b^* Statistically significant difference compared with the ovulation phase in females (*p* = 0.024).

## Data Availability

Data in support of the study findings are available from the corresponding author.

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
