# Peer review of "Relationship between Changes in Foot Arch and Sex Differences during the Menstrual Cycle"

_ijerph, 2022, doi:10.3390/ijerph20010509_

Round 1
Reviewer 1 Report
Dr. Edama investigates the relationship between changes in blood-neutral hormone concentrations during the menstrual cycle and changes in body structure and function continuously.
This research is one of them and the data obtained is very interesting.
There have been several unclear issues, so I have a few questions and suggestions for corrections.
About Exclusion Criteria in Fig.2, suggest for the author's further research.
Several sex hormones, such as E2, are adipocyte-derived. Therefore, extremely underweight subjects may have low levels of E2.
Although not an exclusion criterion for this research, I suggest that authors also consider the exclusion of subjects with low BMI in their future research.
I have a question about statistical methods.
This research compares by sex and menstrual cycle, but have the authors also observed a correlation between sex hormone levels and AHI/AHF in each sex?
Although the sample size is small and the results may not be clear, it may be able to representation the association between sex hormone concentration and AHI/AHF.
Please refer to the author's future research.
In the discussion, the authors describe the possibility of the effect of weight changes during the menstrual cycle (in line 264-275).
Certainly, the AHI requires consideration of weight changes.
On the other hand, when I confirm the calculation method of AHF in line 162, it seem that it is adjusted by body weight.
Is this recognition correct?
The overall impression is that the sample size is small. I would look forward to further development of this research.
Author Response
December 20, 2022
Editorial Board
International Journal of Environmental Research and Public Health
Ref: Submission ID ijerph-2100756
“Relationship between changes in foot arch and sex differences during the menstrual cycle” by Mutsuaki Edama
Dear Editor:
Thank you for your letter. We are grateful for the detailed feedback provided by the reviewers, which we feel has helped us to significantly improve the paper. Attached are our point-by-point responses to the reviewers’ comments and our revised manuscript, which we hope will now meet with your approval. For your convenience, we have attached a copy of the manuscript with all revisions highlighted in red font. We believe that our revisions have addressed the issues raised by the reviewers and trust that the manuscript is now suitable for publication in International Journal of Environmental Research and Public Health.
Thank you again for your thoughtful comments, and we look forward to hearing from you soon.
Sincerely,
Mutsuaki Edama
RESPONSE TO REVIEWER #1
- About Exclusion Criteria in Fig.2, suggest for the author's further research. Several sex hormones, such as E2, are adipocyte derived. Therefore, extremely underweight subjects may have low levels of E2. Although not an exclusion criterion for this research, I suggest that authors also consider the exclusion of subjects with low BMI in their future research.
⇒Thank you for your beneficial comments and suggestions.
In this study, the average BMI of the female subjects was 19.7 ± 2.6. No subjects were extremely low BMI. In the next research, I would like to set the low MBI you pointed out as an exclusion criterion and proceed with the investigation more strictly.
- I have a question about statistical methods. This research compares by sex and menstrual cycle, but have the authors also observed a correlation between sex hormone levels and AHI/AHF in each sex? Although the sample size is small and the results may not be clear, it may be able to representation the association between sex hormone concentration and AHI/AHF. Please refer to the author's future research.
⇒Thank you for your beneficial comments and suggestions.
We as well are interested in the correlation between AHF and E2 concentration and have been analyzing it. However, no significant correlation was observed. We think there is a problem with the small sample size, but I would like to consider it as an issue for future research.
- In the discussion, the authors describe the possibility of the effect of weight changes during the menstrual cycle (in line 264-275). Certainly, the AHI requires consideration of weight changes. On the other hand, when I confirm the calculation method of AHF in line 162, it seems that it is adjusted by body weight. Is this recognition correct?
⇒It is correct. Since AHF is greatly affected by body weight, we focused on the relationship with fluctuations in the menstrual cycle in body weight. Since female athletes are greatly affected not only by weight gain but also by swelling, we would like to investigate the relationship between swelling and AHF in the future.
- The overall impression is that the sample size is small. I would look forward to further development of this research.
⇒At the start of the study, we recruited 67 female subjects, but due to the influence of the COVID-19 and schedule, we ended up with only 10 subjects. In the future, we would like to increase the number of subjects and conduct a more detailed study.

Reviewer 2 Report
I would really appreciate your opportunity to review this manuscript. The concept was very interesting. There are comments to improve the manuscript.
What was your " type of study"?
You used "relationship" in the title, but you did not use any statistical test of relationship in the tour method and you only compared several variables in the two sexes in three stages of the menstrual cycle. would you please explain what was you objectives clearlly?
In the inclusion criteria these were mentioned:
" no orthopedic abnormalities or injuries to the foot and ankle joint; no use of oral contraceptives or other hormonal agents within the past 6 months, no current exercise 82
habit more than twice a week"
It seems these critreries are exclusion critreia. what is your opinion?
What was your sampling method? How did you estimate the sample size?
How did you measure the estradiol and progesterone concentration during the menstrual cycle in you study?
Did you do any maching according to BmI in you groups?
Please mention the scale of variables in the tables.
the discussion part was very short and weak. please re- write this part and mention about the probably causes of no difference between two sexes about the AHI and other results of your stud
Author Response
December 20, 2022
Editorial Board
International Journal of Environmental Research and Public Health
Ref: Submission ID ijerph-2100756
“Relationship between changes in foot arch and sex differences during the menstrual cycle” by Mutsuaki Edama
Dear Editor:
Thank you for your letter. We are grateful for the detailed feedback provided by the reviewers, which we feel has helped us to significantly improve the paper. Attached are our point-by-point responses to the reviewers’ comments and our revised manuscript, which we hope will now meet with your approval. For your convenience, we have attached a copy of the manuscript with all revisions highlighted in red font. We believe that our revisions have addressed the issues raised by the reviewers and trust that the manuscript is now suitable for publication in International Journal of Environmental Research and Public Health.
Thank you again for your thoughtful comments, and we look forward to hearing from you soon.
Sincerely,
Mutsuaki Edama
RESPONSE TO REVIEWER #2
- What was your " type of study"?
⇒The type of study is article (original article).
- You used "relationship" in the title, but you did not use any statistical test of relationship in the tour method, and you only compared several variables in the two sexes in three stages of the menstrual cycle. would you please explain what was your objectives clearlly?
⇒The presence of E2 and P4 receptors in the tissue structures of the foot and ankle joints have not been clarified. However, previous studies have investigated female hormone levels in relation to plantar fascia elasticity and reported that plantar fascia elasticity increases during ovulation, when estrogen levels are maximal. Conversely, a previous study that investigated the rate of anterior talofibular ligament lengthening during the menstrual cycle found no cyclic variation. A clear consensus on the effect of female hormones on foot posture thus remains lacking.
The correlation between arch structure and injury may be related to the fact that foot structure influences foot function. Foot structure is often defined by arch height, although arch flexibility may be just as important to form a more complete description. A previous study found that runners with high arch structure but differing arch mobility exhibited differences in select lower extremity movement pat-terns and forces. Evaluating AHI and AHF is thus considered useful for clarifying the relationship between foot misalignment and foot injury. A previous study that evaluated foot posture using AHI and AHF reported that AHI was significantly lower in women than in men under all loading conditions (10%, 50%, and 90%), and AHF decreased with increasing loading in both sexes. However, the effects of fluctuations in female hormones during the menstrual cycle on changes in AHI and AHF remain unclear.
Therefore, in the present study, to clarify the relationship between foot posture changes and sex differences during the menstrual cycle (early follicular phase, ovulatory phase, and luteal phase) in healthy male and female university students. We hypothesized that AHF would be significantly higher in the ovulatory and luteal phases of the menstrual cycle in females than in the early follicular phase.
- In the inclusion criteria these were mentioned: " no orthopedic abnormalities or injuries to the foot and ankle joint; no use of oral contraceptives or other hormonal agents within the past 6 months, no current exercise 2 habit more than twice a week" It seems these critreries are exclusion critreia. what is your opinion?
⇒Considering the inclusion criteria based on previous research. This is also an inclusion criterion in our previous studies 1‐3).
1)Shagawa, M. Maruyama, S. Sekine, C. Yokota, H. Hirabayashi, R. Hirata, A. Yokoyama, M. Edama, M. Comparison of anterior knee laxity, stiffness, genu recurvatum, and general joint laxity in the late follicular phase and the ovulatory phase of the menstrual cycle. BMC musculoskeletal disorders 2021, 22, 886, doi:10.1186/s12891-021-04767-8.
2)Yamazaki, T. Maruyama, S. Sato, Y. Suzuki, Y. Shimizu, S. Kaneko, F. Ikezu, M. Matsuzawa, K. Edama, M. A preliminary study exploring the change in ankle joint laxity and general joint laxity during the menstrual cycle in cis women. Journal of foot and ankle research 2021, 14, 21, doi:10.1186/s13047-021-00459-7.
3)Maruyama S, Yamazaki T, Sato Y, Suzuki Y, Shimizu S, Ikezu M, Kaneko F, Matsuzawa K, Hirabayashi R, Edama M. Relationship Between Anterior Knee Laxity and General Joint Laxity During the Menstrual Cycle. Orthop J Sports Med. 2021. 29;9(3):2325967121993045. doi: 10.1177/2325967121993045.
- What was your sampling method? How did you estimate the sample size?
⇒When designing the study, we estimated a sample size of 20 people. Therefore, at the start of the study, we recruited 67 female subjects, but due to the influence of the COVID-19 and schedule, we ended up with only 10 subjects. In the future, we would like to increase the number of subjects and conduct a more detailed study.
- How did you measure the estradiol and progesterone concentration during the menstrual cycle in you study?
⇒Detailed methods are described in 2.4. Measurement methods (Lns 126-143).
2.4. Measurement methods
Using a saliva collection kit (SalivaBio A; Salimetrics, Carlsbad, CA, USA), E2 and P4 concentrations were measured. Subjects strictly obeyed the following points before saliva collection to avoid possible influences on E2 and P4 concentrations: 1) Food intake re-strictions within 60 minutes; 2) dairy product restrictions within 20 minutes; 3) alcohol restrictions within 12 hours; 4) sugary, acidic, or caffeinated drink restrictions within 20 minutes; and 6) saliva collection restrictions during 48 hours after dental treatment. Additionally, participants were instructed to rinse their mouths before the experiment began to remove any food particles. Saliva samples were also taken more than 10 minutes after mouthwash to prevent decreases in E2 concentration. A straw (Siva Collection Aid; SAL) was used to collect saliva in the mouth for one minute before it was drained into a saliva collection container (Cryovial; SAL). Immediately following collection, the saliva sample was frozen in a freezer at below - 80 °C. After gathering all samples, Funakoshi Corporation (Tokyo, Japan) was entrusted with the analysis of E2 concentrations. The High Sensitivity Salivary 17-Estradiol Enzyme Immunoassay Kit (SALIMETRICS) was used to measure the E2 concentrations. Samples were thawed at room temperature, mixed by vertexing, centrifuged at 1500 g for 15 min., and then analyzed using an enzyme-linked immunosorbent assay. The dilutions were all consistently onefold (undiluted solution) [23,24].
23)Shagawa, M.; Maruyama, S.; Sekine, C.; Yokota, H.; Hirabayashi, R.; Hirata, A.; Yokoyama, M.; Edama, M. Comparison of anterior knee laxity, stiffness, genu recurvatum, and general joint laxity in the late follicular phase and the ovulatory phase of the menstrual cycle. BMC musculoskeletal disorders 2021, 22, 886, doi:10.1186/s12891-021-04767-8.
24)Maruyama, S.; Sekine, C.; Shagawa, M.; Yokota, H.; Hirabayashi, R.; Togashi, R.; Yamada, Y.; Hamano, R.; Ito, A.; Sato, D., et al. Menstrual Cycle Changes Joint Laxity in Females-Differences between Eumenorrhea and Oligomenorrhea. J Clin Med 2022, 11, doi:10.3390/jcm11113222.
- Did you do any maching according to BmI in your groups?
⇒Does BmI mean body mass index? In this study, the average BMI of the female subjects was 19.7 ± 2.6. No subjects were extremely low BMI. In the next research, I would like to set the low MBI you pointed out as an exclusion criterion and proceed with the investigation more strictly.
- Please mention the scale of variables in the tables.
⇒The table 2 has been corrected as suggested.
Table 2. Changes in AHI and AHF during the menstrual cycle
|
Early follicular phase (Phase 1) |
Ovulation phase (Phase 2) |
Luteal phase (Phase 3) |
Total |
Body Weight (Kg) |
|
|
|
|
Male (n = 14) |
62.7 ± 7.3 |
62.8 ± 7.0 |
62.8 ± 7.0 |
62.7 ± 7.1 |
Female (n = 10) |
50.4 ± 8.2 |
50.2 ± 7.8 |
50.4 ± 7.9 |
50.3 ± 8.0 |
AHI (10% load condition) (unitless) |
|
|
|
|
Male (n = 14) |
0.38 ± 0.02 a |
0.38 ± 0.02 b |
0.38 ± 0.01 |
0.38 ± 0.02 |
Female (n = 10) |
0.37 ± 0.02 |
0.37 ± 0.02 |
0.37 ± 0.02 |
0.37 ± 0.02 |
AHI (50% lord condition) (unitless) |
|
|
|
|
Male (n = 14) |
0.35 ± 0.01 |
0.35 ± 0.02 |
0.35 ± 0.02 |
0.35 ± 0.02 |
Female (n = 10) |
0.34 ± 0.02 |
0.34 ± 0.02 |
0.34 ± 0.02 |
0.34 ± 0.02 |
10–50%AHF (mm/kN) |
|
|
|
|
Male (n = 14) |
19.6 ± 4.3 |
20.9 ± 5.3 |
18.7 ± 5.4 |
19.7 ± 5.0 |
Female (n = 10) |
19.2 ± 5.6 |
21.2 ± 5.2 |
22.4 ± 5.9 |
19.6 ± 4.3 |
Values are presented as mean ± SD.
aStatistically significant difference compared with the early follicular phase in females (P = .032).
bStatistically significant difference compared with the ovulation phase in female (P = .024).
- The discussion part was very short and weak. please rewrite this part and mention about the probably causes of no difference between two sexes about the AHI and other results of your study.
⇒The text has been corrected as suggested. (Lns 250-271)
In this study, AHI (10% load condition) was significantly lower in females than in males in the early follicular and ovulatory phases, but no significant difference was seen AHI (50% load condition). There was no significant difference between menstrual cycles in both AHI. AHF did not differ significantly between males and females in each phase. Previous studies have investigated AHI in all load conditions (10%, 50%, and 90%of weight bearing load) showed significant differences between the genders, and the AHI of female participants was significantly less than that of male participants [12]. Although the results were different from the results of this study, previous study [12] used the arch height index measurement system to measure AHI, but in this study, a 3-dimensional foot scanner was used. It was suggested that the difference might have an effect. In addition, previous studies have investigated female hormone levels in relation to plantar fascia elasticity, and reported that plantar fascia elasticity increases during ovulation, when estrogen levels are maximal [10,11]. On the other hand, a previous study [11] that investigated the rate of anterior talofibular ligament lengthening during the menstrual cycle reported no cyclic variation. Thus, a clear, consistent view on the effect of female hormones on foot posture remains lacking. In this study, E2 concentrations were significantly higher in the ovulatory phase than in the early follicular phase. P4 concentration was significantly higher in the luteal phase than in the early and late follicular phases, suggesting that E2 and P4 concentrations showed normal cyclic variation. Although the presence or absence of E2 and P4 receptors in the foot tissue structure has not been clarified [10,11], AHI and AHF, which indicate foot characteristics in the menstrual cycle, may not be affected by E2 and P4 and may not fluctuate cyclically.
10) Petrofsky, J.; Lee, H. Greater Reduction of Balance as a Result of Increased Plantar Fascia Elasticity at Ovulation during the Menstrual Cycle. The Tohoku journal of experimental medicine 2015, 237, 219-226, doi:10.1620/tjem.237.219.
11) Yamazaki, T.; Maruyama, S.; Sato, Y.; Suzuki, Y.; Shimizu, S.; Kaneko, F.; Ikezu, M.; Matsuzawa, K.; Edama, M. A preliminary study exploring the change in ankle joint laxity and general joint laxity during the menstrual cycle in cis women. Journal of foot and ankle research 2021, 14, 21, doi:10.1186/s13047-021-00459-7.
12) Takabayashi, T.; Edama, M.; Inai, T.; Nakamura, E.; Kubo, M. Effect of Gender and Load Conditions on Foot Arch Height Index and Flexibility in Japanese Youths. J Foot Ankle Surg 2020, 59, 1144-1147, doi:10.1053/j.jfas.2020.03.019.

Round 2
Reviewer 2 Report
Thanks for your response
I seems your study type is " Analytic - observational cross- sectional"
Author Response
December 25, 2022
Editorial Board
International Journal of Environmental Research and Public Health
Ref: Submission ID ijerph-2100756
“Relationship between changes in foot arch and sex differences during the menstrual cycle” by Mutsuaki Edama
Dear Editor:
Thank you for your letter. We are grateful for the detailed feedback provided by the reviewers, which we feel has helped us to significantly improve the paper. Attached are our point-by-point responses to the reviewers’ comments and our revised manuscript, which we hope will now meet with your approval. For your convenience, we have attached a copy of the manuscript with all revisions highlighted in red font. We believe that our revisions have addressed the issues raised by the reviewers and trust that the manuscript is now suitable for publication in the International Journal of Environmental Research and Public Health.
Thank you again for your thoughtful comments, and we look forward to hearing from you soon.
Sincerely,
Mutsuaki Edama
RESPONSE TO REVIEWER #2
- It seems your study type is " Analytic - observational cross-sectional"
⇒The text has been corrected as suggested.
This study type is changed from " Article" to " Analytic - observational cross-sectional"